# High Flexibility of RNaseH2 Catalytic Activity with Respect to Non-Canonical DNA Structures

**DOI:** 10.3390/ijms22105201

**Published:** 2021-05-14

**Authors:** Maria Dede, Silvia Napolitano, Anna Melati, Valentina Pirota, Giovanni Maga, Emmanuele Crespan

**Affiliations:** 1Institute of Molecular Genetics IGM-CNR “Luigi Luca Cavalli-Sforza”, via Abbiategrasso 207, I-27100 Pavia, Italy; maria.dede01@universitadipavia.it (M.D.); silvia.napolitano@mol.biol.ethz.ch (S.N.); anna.melati01@universitadipavia.it (A.M.); giovanni.maga@igm.cnr.it (G.M.); 2Chemistry Department, University of Pavia, Via Taramelli 12, I-27100 Pavia, Italy; valentina.pirota@unipv.it

**Keywords:** RNaseH2, misincorporated ribonucleotides, RER, non-B DNA

## Abstract

Ribonucleotides misincorporated in the human genome are the most abundant DNA lesions. The 2′-hydroxyl group makes them prone to spontaneous hydrolysis, potentially resulting in strand breaks. Moreover, their presence may decrease the rate of DNA replication causing replicative fork stalling and collapse. Ribonucleotide removal is initiated by Ribonuclease H2 (RNase H2), the key player in Ribonucleotide Excision Repair (RER). Its absence leads to embryonic lethality in mice, while mutations decreasing its activity cause Aicardi–Goutières syndrome. DNA geometry can be altered by DNA lesions or by peculiar sequences forming secondary structures, like G-quadruplex (G4) and trinucleotide repeats (TNR) hairpins, which significantly differ from canonical B-form. Ribonucleotides pairing to lesioned nucleotides, or incorporated within non-B DNA structures could avoid RNase H2 recognition, potentially contributing to genome instability. In this work, we investigate the ability of RNase H2 to process misincorporated ribonucleotides in a panel of DNA substrates showing different geometrical features. RNase H2 proved to be a flexible enzyme, recognizing as a substrate the majority of the constructs we generated. However, some geometrical features and non-canonical DNA structures severely impaired its activity, suggesting a relevant role of misincorporated ribonucleotides in the physiological instability of specific DNA sequences.

## 1. Introduction

Ribonucleotides embedded in the genome are the most abundant lesions in living organisms. More than 10^7^ ribonucleotides are misincorporated in the human genome by DNA polymerases (Pols) during replication or DNA repair pathways [1,2,3]. It has been suggested that ribonucleotide incorporation could assist the recognition of the newly synthesized DNA strand by the mismatch repair machinery [4,5]. However, they can also constitute a major threat to genome stability [6,7]. The sugar-phosphate backbone of RNA is much more prone to strand breakage than DNA, potentially resulting in the accumulation of strand breaks [1]. Moreover, if not promptly removed, accumulation of misincorporated ribonucleotides may decrease the rate of DNA replication [3,8,9,10], causing replicative fork stalling and, eventually, collapse, resulting in single and double strand breaks. To avoid these issues, misincorporated ribonucleotides are removed by the Ribonucleotide Excision Repair pathway (RER), which is initiated by Ribonuclease H2 (RNase H2), which incises the phosphodiester bond at the 5′ of the embedded ribonucleotide(s) in the hybrid filament [11]. RNase H2 is a very specialized enzyme extremely conserved throughout prokaryotic and eukaryotic organisms and its activity is essential in eukaryotic cells. In yeast, mutations in RNaseH2 cause replication stress, requiring translesion synthesis (TLS) to avoid ribonucleotide accumulation in the genome [12]. In mice, the absence of RNase H2 leads to embryonic lethality, while in humans, the severe phenotype of the Aicardi–Goutières syndrome is due to mutations in RNaseH2 [13].

We recently evaluated the ability of human RNase H2 to incise ribonucleotides paired to DNA-damaged bases [14]. We found that RNase H2 works slightly less efficiently on ribo C (rC) paired to 8-oxoguanine (8-oxo-G), the major oxidative damage in the genome, with respect to rC paired with a canonical dG. Furthermore, RNase H2 processes, with almost the same efficiency, substrates containing an rC paired with a canonical dG, or with the alkylated base 8-methyl-2-deoxyguanosine (8-met-G). On the other hand, we observed that a single ribonucleotide paired to a cisplatin adduct (cis-Pt) is a poor substrate for RNase H2. 8-oxo-G and 8-met-G, while altering the pairing properties of the nucleobase, do not substantially disrupt the geometrical structure of the double helix. On the contrary, Cis-Pt distorts the geometry of the double helix and decreases its flexibility. Other lesions, such as cyclobutane pyrimidine dimers, are able to distort the canonical B-helix geometry. Moreover, peculiar sequences within the genome can trigger the formation of secondary structures such as G-quadruplex (G4) and stem-loop that are referred collectively to as non-B DNA structures [15]. Such lesions and structures are associated with genomic instability and represent a challenge for the progression of transcription and replication forks. The presence of misincorporated ribonucleotides in such structures could represent an additional threat to replication fork progression.

In this work, we designed a panel of DNA substrates showing different geometrical features and bearing a single embedded ribonucleotide at specific positions, to dissect the ability of human RNase H2 to recognize and incise ribonucleotides in non-canonical DNA structures. RNase H2 proved to be a flexible enzyme, able to recognize as substrate the majority of the constructs we generated. However, some geometrical features severely impaired the activity of the enzyme, suggesting a relevant role of misincorporated ribonucleotides in the physiological instability of peculiar DNA sequences.

## 2. Results

### 2.1. A DNA Double Stranded Structure at the 3′ of the Ribonucleotide Is Not Required for RNase H2 Activity

To understand the geometric properties required by RNase H2 to recognize and process a DNA/RNA hybrid composed of a single embedded ribonucleotide, we generated a set of constructs with a common hybrid strand bearing a single rC that was paired with complementary DNA strands, creating different secondary structures. Time-course experiments were performed in parallel using a fully double-stranded substrate (Substrate **1**) as the reference canonical substrate.

First, RNase H2 was tested in a time-course experiment on a partially double-stranded DNA substrate, with the unpaired region flanking the 3′ of the embedded rC (Substrate **2**). The fully paired substrate (Substrate **1**) was used as a control. Under these conditions, RNase H2 activity was comparable with both substrates (Figure 1B, compare lanes 2–6 with lanes 8–12, Appendix A), proving that strand pairing at the 3′ of the embedded ribonucleotide was not essential for enzyme activity. However, when the ribonucleotide rC was included in the flap structure (Substrates **3** and **S1**), the enzyme activity was almost completely abolished (Figure 1C, compare lanes 2–6 with lanes 8–12, Appendix A).

### 2.2. RNase H2 Can Process a Mismatched Ribonucleotide Embedded within an Intact DNA Double-Stranded Structure

The lack of activity on Substrate 3 (Figure 1C and Appendix A) could reflect the inability of RNase H2 to process a mismatched ribonucleotide independently in the presence of a single- or double-stranded DNA flanking its 3′. To verify this possibility, RNase H2 was tested on a substrate in which the rC formed a mismatch with the opposite nucleotide (rC:T) but was flanked by correctly paired regions either at its 3′ and 5′ (Substrate **4**). Under these conditions, RNaseH2 could efficiently process the mismatched rC (Figure 1D, compare lanes 2–6 with lanes 8–12; Appendix A), indicating that mispairing of the ribonucleotide is not important for the activity of RNase H2. Notably, the reaction efficiency on Substrate **4** was higher than on Substrate **1** (Figure 1D,E). The same result was obtained with rC:A mismatch (Substrate **S2**, Appendix A).

### 2.3. RNase H2 Can Process Ribonucleotides at the Junction of Flap-Like Structures at the 3′ of the Embedded Ribonucleotide

Annealing of a 21mer oligonucleotide to Substrates **2** and **3** generated Substrates **5** and **6**, respectively. Such substrates mimic structures that could originate from the strand displacement activity of DNA pols that stalled when encountering a ribonucleotide on the template strand. As expected, the presence of a flap-like structure flanking the 3′ of a correctly paired rC (Substrate **5**) did not affect RNase H2 activity (Figure 1F, compare lanes 2–6 with lanes 8–12). Interestingly, the presence of a double-stranded region at the 3′ of the rC restored the activity of the enzyme even when the ribonucleotide was not annealed to the template and included in the flap double-stranded region at the 3′ of the rC restored the activity of the enzyme (Figure 1G, compare lanes 2–6 with lanes 8–12).

### 2.4. Base Pairing at the 5′ of the Embedded Ribonucleotide Is Essential for RNase H2 Activity

While unpaired strands downstream of the embedded nucleotide were not essential for RNase H2 activity, the presence of a proper paired structure at 5′ was essential for the reaction. In particular, a single-stranded structure flanking rC at its 5′ (Substrate **7**) completely abolished the reaction (Figure 2B and Appendix A). Interestingly, enzyme activity was restored when a short oligonucleotide covering the single-stranded region was annealed to the rC-containing strand, generating a flapped structure (Substrate **8**, Figure 2C and Appendix A).

To understand the length of the double-stranded tract at the 5′ of the ribonucleotide required for RNase H2 activity, we generated three substrates presenting a 5′ flap and 5, 2, or 1 paired nucleotides upstream the rC (Substrates 9, 10, and 11, respectively, see Figure 2A). Full enzymatic activity was observed in the presence of five paired nucleotides upstream of the ribonucleotide; this was sufficient to completely restore the enzymatic activity (Figure 2D, compare lanes 2–6 with lanes 7–11; Appendix A). Pairing of the first two nucleotides flanking rC (Substrate 10) almost rescued completely RNase H2 activity, especially at longer time points (Figure 2E, compare lanes 2–6 with lanes 8–12; Appendix A) while, not surprisingly, the pairing of the solely first nucleotide flanking the rC before the flapped region (Substrate 11) resulted in a loss of more than 50% of enzyme activity (Figure 2F compare lanes 2–6 with lanes 8–12; Appendix A).

### 2.5. DNA Mismatches at the 5′ of the Ribonucleotide Differentially Affects RNase H2 Activity

We further evaluated the relevance of a correctly paired double-stranded structure flanking the 5′ of the embedded ribonucleotide using a set of substrates presenting one or more mismatched base pairs. The presence of a single mismatch immediately upstream (position -1) from the ribonucleotide (Substrate **12**) clearly impaired RNase H2 activity (Figure 3B, compare lanes 2–6 with lanes 8–12; Appendix A). When the mismatch involved also position -2, that is the second base pair further upstream (Substrate **13**), the activity of the enzyme was almost completely lost (Figure 3C compare lanes 2–6 with lanes 8–12; Appendix A). However, when a mismatch involved positions -2 and -3 upstream from the ribonucleaotide, but not position -1 (Substrate **14**), RNase H2 activity was less affected (Figure 3D compare lanes 2–6 with lanes 8–12; Appendix A). This indicates that the presence of a correct base pairing of the nucleotides immediately flanking the ribonucleotide at its 5′ allowed RNaseH2 to tolerate even two mismatched nucleotides further upstream. Next, RNase H2 was challenged with a more extreme case, where the nucleotides at positions -2 and -3 at the 5′ of the rC form two mismatches, the position -1 proximal to the 5′ of the rC is correctly paired, but the rC itself pairs with a T on the complementary strand, forming an additional mismatch (Substrate **15**). On this substrate, the sole annealing of the -1 base pair flanking the 5′ of embedded mismatched ribonucleotide was sufficient for RNase H2 activity, even with a slightly reduced efficiency with respect to the canonical Substrate **1** (Figure 3E compare lanes 2–6 with lanes 8–12; Appendix A).

All together, these results indicate that RNase H2 possesses the ability to recognize and process ribonucleotides in the context of various secondary structures, but it requires proper base pairing of at least one nucleotide flanking the 5′ of the embedded ribonucleotide.

### 2.6. Non-B DNA Structures Affect RNase H2 Activity

Next, we evaluated the activity of RNase H2 on non-B DNA structures that naturally occur in physiological conditions.

A single strand presenting a guanine rich sequence can fold in a four-stranded structure called G-quadruplex (G4). In more details, four guanine bases associate through Hoogsteen hydrogen bonding, forming square planar structures (G-quartet) that form a pile, stacked on top of each other. These secondary structures are helical in shape and could occur naturally at telomeric regions or in transcriptional regulatory regions of multiple genes, including several oncogene promoters [16]. Their formation is promoted when single-stranded regions are exposed, such as during transcription and replication. G4 are extremely stable structures that could compete with canonical double strand DNA conformation [17].

RNaseH2 was proved to efficiently process substrates presenting a single-stranded region immediately flanking the 3′ of an embedded ribonucleotide. On the other hand, the integrity of a proper double-stranded structure at the 5′ of the ribonucleotide was essential for the reaction. Therefore, we decided to evaluate the activity of the enzyme on substrates presenting two different sequences at the 3′ of a single embedded ribonucleotide, both known to form stable G4 structures (GGGATAGGGATAGGGATAGGG sequence within Substrate **16** and cmyc22 sequence—TGAGGGTGGGTAGGGTGGGTAA [18] within Substrate **S3**). The formation of G4 structure was achieved by annealing the proper oligonucleotides in the presence of K^+^ ions, using control oligonucleotides annealed in presence of Li^+^ ions (see materials and methods). G4 formation was confirmed by CD spectroscopy and melting analysis (see Appendix A) [18,19,20,21]. In Substrates **16** and **S3**, the sequence opposite to the G4 consensus motif in the DNA strand, consisted of a stretch of adenosines, avoiding any competition between G4 formation and canonical interstrand annealing (Figure 4A). As expected, the presence of a single-stranded region at the 3′ of the ribonucleotide did not impair enzyme activity. On the other hand, the presence of the G4 structure strongly impaired RNase H2 activity. (Figure 4B and Appendix A, compare lanes 2–6 with lanes 8–12). In the presence of K^+^, Substrate **S4** forms a G4 structure at the 5′ of the embedded ribonucleotide, thus disrupting proper annealing at the 5′ of the embedded ribonucleotide and completely impairing the enzyme activity, which is completely restored when the absence of K^+^ allows proper strands annealing (Appendix A, compare lane 3 with lane 1). Since RNase H2 proved to efficiently process substrates with the unpaired region flanking the 3′ of an embedded ribonucleotide, we designed Substrate **17** that presents a G4 consensus sequence (GGGATAGGGATAGGGATAGGG) at the 3′ of the ribonucleotide but on the DNA strand. Despite the enzyme showing a higher activity when the substrate was annealed in the absence of K^+^, substantial activity was observed also in the presence of the G4 structure.

It is believed that long tracts composed by tandem trinucleotide repeats (TNR) could undergo conformational changes resulting in thermodynamically stable stem-loop structures formation. The stems of such structures consist of the repeated pattern of two paired nucleotides followed by a single nucleotide mismatch. It is believed that the formation of such structures is at the base of the intrinsic genomic instability of such sequences. For instance, long (CAG) trinucleotides repeats are associated with triplets expansion in neurodegenerative disorders, while (CCG) repeats are linked to chromosome fragility in folate sensitive rare fragile sites [22].

To evaluate the ability of RNase H2 to process an embedded ribonucleotide within such structures, we constructed three substrates formed by just three CCG repeats, mimicking the stem part of the hairpin. In each substrate, a ribonucleotide was inserted in a different position within the central triplet (Figure 5A, Substrates **19**, **21**, and **23**). As controls, the same sequences were annealed with a fully complementary strand, resulting in canonical double stranded oligonucleotides (Figure 5A, Substrates **18**, **20**, and **22**). The difference in the folding of the structure was highlighted by CD spectra, comparing Substrates **18** and **19**. The CD spectrum of Substrate **18** has a maximum at 279 nm and a minimum at 246 nm (Appendix A, dashed grey line), typical features of a B-DNA structure. Likewise, the signal of Substrate **19** presents a maximum at 280 nm and a minimum at 248 nm, but showing a slight shift to higher wavelengths with a CD amplitude reduction (Appendix A, bold black line), congruent with the duplex to hairpin transition [19,23].

Overall, the stem structure strongly affects RNase H2 activity, as can be observed from the reduction of products accumulation with respect toward control substrates (Figure 5B,C, compare lanes 2–6 with lanes 8–12). Moreover, the position of the ribonucleotide within the triplet modulated the enzyme reaction. In fact, according to what was observed with Substrate **12** and Substrate **4**, the presence of a mispair at the 5′ of the ribonucleotide (Substrate **23**) affected the reaction more severely (Figure 5E).

### 2.7. RNase H2 Activity Is Not Affected by Cyclobutane Thymine Dimers Opposite the Embedded Ribonucleotide

In a previous work [14], we have shown that RNaseH2 was able to remove a rC paired to the first (3′) G of a Cis-Pt G-G adduct. To assess whether other helix distorting lesions affected the ability of RNaseH2 to remove the misincorporated ribonucleotides, a substrate containing an rA pairing with the first T of a cyclobutane thymine dimer (CTD) was tested (Substrate **24**). CTDs consist of thymine or cytosine bases crosslinked via photochemical reactions catalyzed by UV light and represent a common DNA lesion, requiring the specialized DNA pol η for TLS, as in the case of Cis-Pt adducts. During TLS, DNA pol η can misincorporate ribonucleotides, hence the interest in assessing whether RNase H2 could be able to clean up such ribonucleotides after TLS occurred. As shown in Figure 6, RNase H2 was able to process rA paired to a CTD with the same efficiency showed toward an undamaged substrate (Substrate **25**).

## 3. Discussion

To explore the ability of human RNase H2 to process substrates with different geometrical properties, we constructed a series of oligonucleotides bearing a single embedded ribonucleotide and mismatched nucleotides in different positions, as well as single stranded regions. We discovered that RNase H2 is able to recognize and process substrates bearing mismatched nucleotides. In particular, RNase H2 incises more efficiently the substrate when the ribonucleotide itself forms a mismatch with the complementary base (Figure 1C,D and Appendix A). This is not surprising since, in *E. coli* and *S. cerevisiae*, oligonucleotide-driven gene correction assays revealed that single mismatched ribonucleotides are efficiently correct in the presence of RNase HII [24]. Furthermore, it is thought that RNase H2 recognizes the larger minor groove in the double helix formed by RNA/DNA hybrids. Mismatches in RNA/DNA would further increase the minor groove width, favoring RNase H2 recognition. In addition, based on the structure of the RNaseH2 from *T. maritima* in complex with its substrate, it has been proposed that the interaction of the active site of the enzyme with the sugar-phosphate backbone in the vicinity of the ribonucleotide causes a deformation of the 5′-RNA-DNA-3′ junction, facilitating cleavage [25]. The presence of a mismatched base pair lacking Watson–Crick hydrogen bonds might exacerbate such a distortion, thereby increasing the enzyme efficiency.

We also found that the presence of a canonical double-stranded structure at the 3′ of the embedded ribonucleotide is not required for enzyme activity, which is unaffected by the presence of unpaired strands (Figure 1C), as well as by the presence of flapped structures (Figure 1F,G) flanking the ribonucleotide.

These structures are generated in vivo by strand displacement activity of DNA polymerases both during DNA replication and in some repair events. They are removed by flap endonuclease 1 (FEN1) activity under the coordination of PCNA. Flaps removal is particularly relevant during Okazaki fragment maturation, in which RNA primers are displaced by pol δ and removed by the combined activity of RNase H1 or RNase H2 and FEN1 [26]. A similar scenario is proposed to occur in RER, where RNase H2 initiates the process incising the DNA/RNA junction at the 5′ of the embedded ribonucleotide(s). Next, a DNA polymerase extends the nick forming a flap that is incised by FEN1 [11]. Therefore, a flapped structure could represent natural substrates for RNase H2.

The only available crystal structure of RNase H2 in complex with a DNA/RNA hybrid substrate was obtained with *T. maritima* [25]. The enzyme was bound to a double-stranded oligonucleotide presenting a hybrid DNA oligonucleotide with a single ribonucleotide (hybrid strand), paired to a fully DNA strand (DNA strand). While no residues interacted with the DNA strand at the 3′ of the ribonucleotide (downstream), the structure revealed a network of interactions with the four nucleotides in the DNA strand at the 5′ (upstream) of the embedded ribonucleotide. Accordingly, the presence of unpaired regions upstream of the embedded ribonucleotide completely abolishes RNase H2 activity, but the correct pairing of the two base pairs at the 5′ of the ribonucleotide almost completely restores the activity of the enzyme (Substrate **10**, Figure 2E) which is completely rescued when the base pair tract involves 5 base pair upstream of the ribonucleotide (Substrate **9**). In *T. maritima*, R22 (R38 in human RNase H2A subunit) is a key residue for the catalytic activity. Through a water molecule interaction, it stabilizes the second nucleotide at the 5′ of the ribonucleotide in the DNA strand. Together with G21 and G23, R22 constitutes the highly conserved GRG motive that is involved in the recognition of the RNA/DNA transition [25]. The abrogation of R22 interaction with the DNA strand may influence the proper conformation of the GRG motive, thus impairing the catalytic activity.

G4 structures are prone to form at CpG islands in gene promoters as well as G-rich telomeric sequences, where their formation has been proposed to play relevant biological roles [27]. Incorporation of ribonucleotides within these sequences could alter their predisposition to acquire a G4 structure, and thus the ability of RNase H2 to remove ribonucleotides within these structures could be physiologically relevant. Our data showed that RNase H2 is not able to cleave a ribonucleotide flanking a G4 on the same strand. However, the presence of a G4 in the DNA strand at the 3′ of the ribonucleotide only modestly reduces RNase H2 activity (Figure 4C and Appendix A). This is in agreement with the ability of RNase H2 to efficiently process substrates showing unpaired regions flanking the 3′ of the embedded ribonucleotide.

Oligonucleotides resembling the stem part of hairpin structures formed by CCG TNR have been shown to be poor substrates of RNase H2. Furthermore, in this case, RNase H2-impaired activity could be due to the presence of single mismatches at the 5′ of the ribonucleotide. These involved the first, second, and third base pair from the ribonucleotide (Substrates **23**, **19**, and **21**, respectively). Accordingly, the catalytic activity was particularly affected by Substrate **23**, where the mismatch was proximal to the embedded ribonucleotide. Nonetheless, this explanation does not fully justify the strongest impairment of the reaction observed with these substrates compared to those presenting mismatches within a random sequence. For instance, the reaction was less efficient at processing Substrate **22** than Substrate **12**, even if both present a mismatch of the base pair proximal to the ribonucleotide (compare Figure 3B with Figure 5D). In the same way, RNase H2 was more active on Substrate **14**, which exhibits a mismatch involving the second and the third base pair at the 5′ of the ribonucleotide, than on Substrate **19**, where the mismatch at the 5′ of the ribonucleotide involves only the second base pair (compare Figure 3D with Figure 5B). However, in the case of Substrates **22** and **12**, it is important to note that the presence of a different base (rG and rC, respectively) could influence the efficiency of the reaction. Using fully paired substrates, we previously showed that RNase H2 incised the rG:dC pair 2,5 fold less efficiently than rC:dG [14], potentially contributing to the reduced activity observed on Substrate **22**. Despite hairpins constituted by CNG-repeated sequences assuming a double-stranded helical conformation, the repetitive pattern of mismatched base pairs alters the canonical geometrical conformation of the DNA that, within these structures, can assume a non-B helix geometry such as Z-type helix [28]. For these reasons rather than the position of the embedded ribonucleotide with respect to the G-G mispair, the overall structure of the CGG hairpin could account for the majority of the loss of RNase H2 activity.

TNR are known to be regions difficult to replicate and to be hotspots for genomic instability. In particular, expanded CGG repeated tracts are typical of folate-sensitive rare fragile sites. Our data suggest that RNase H2 poorly processed ribonucleotides misincorporated in hairpin structures formed within these regions. It is possible to speculate that unrepaired misincorporated ribonucleotides would represent a further obstacle to fork progression, favoring fork collapse and, ultimately, DNA breaks, thus contributing to an explanation of the genetic instability of these regions. Importantly, our data do not exclude that the low RNase H2 activity observed toward non-B DNA substrates would arise from a decreased binding affinity rather than from cleavage efficiency. In vivo, unrevealed factors favoring association of RNase H2 to its substrates would increase its efficiency. Future studies evaluating the binding affinity of RNase H2 toward these substrates may clarify this point.

DNA polymerase η (pol η), the enzyme devoted to the TLS of naturally occurring CTDs and for the bypass of Cis-Pt adducts [29], intriguingly exhibits a high rate of ribonucleotide misincorporation [14,30]. Both CTDs and Cis-PT adducts introduce rigidity to the double helix, distorting its geometry. However, our results suggest that misincorporation of ribonucleotides toward these lesions does not impair RNase H2activity (see. Mentegari et al. [14] and Figure 6).

In conclusion, our results show that RNase H2 is able to incise both substrates that differ from the canonical double stranded conformation, as well as ribonucleotides pairing with several DNA lesions [31]. However, it is known that in some contexts, RNase H2 is not able to initiate the ribonucleotide removal. It is the case of oxidized and abasic ribonucleotides that are instead incised by the Base Excision Repair enzyme AP1 [32,33]. In *E. coli*, in the absence of RNase HII, MMR and NER have been shown to play a role in ribonucleotide removal in bacteria [24,34]. Within the cell, the misincorporation of ribonucleotides in regions forming non-B DNA structures such as G4 or TNR hairpins is not a negligible event. It is, therefore, possible to hypothesize that specialized DNA repair pathways, may be based on the so far unexplored activity of known DNA repair factors which play a role in the repair of these lesions.

## 4. Materials and Methods

### 4.1. DNA Substrates

All oligonucleotides were HPLC purified and purchased from Biomers.net GmbH (Ulm, Germany). Sequences used for the generation of the different substrates are listed in Table 1. Where indicated, oligonucleotides were directly synthesized as 5′-labelled with carboxyfluorescein (Fam) group. Each substrate was obtained by mixing the proper oligonucleotides at a 1:1 (M/M) ratio in annealing buffer (30 mM HEPES KOH pH 7.4, 100 mM KCl, 2 mM MgCl2, and 50 mM NH4Ac), heated at 100 °C for 10 min and then slowly cooled down to room temperature. To avoid the generation of a G-quadruplex structure (Substrates **16** and **17**), oligonucleotides #17–#18 and #19–20# were mixed into a modified annealing buffer in which 100 mM KCl was substituted with 100 mM LiCl and HEPES KOH with HEPES LiCl. If unspecified, all reagents come from AppliChem (Darmastadt, Germany).

### 4.2. RNase H2 Purification

The MIC1066 *E. coli* strain and the expression plasmid for the simultaneous expression of the three subunits of human RNase H2 were kindly provided by R. J. Crouch (National Institute of Child Health and Human Development, Bethesda, MD, USA). The heterotrimeric human recombinant RNase H2 was expressed and purified as described [14,34], with the following modifications: cells were lysed for 30 min on ice in 20 mM KPO4 pH 7.4, 300 mM NaCl, 30 mM Imidazole, 10 mg/mL lysozyme, 0.05% phenylmethylsulfonyl fluoride and protease inhibitors cocktail (Sigma-Aldrich, Munich, Germany). The lysate was sonicated and cleared by ultracentrifugation at 38,000 rpm in a Ti70 Beckman rotor for one hour. The cleared lysate was loaded onto an FPLC-NiNTA column (GE Healthcare, Chicago, Illinois, USA) equilibrated in 20 mM KPO4 pH 7.4, 300 mM NaCl, 30 mM imidazole, and 5% glycerol. The protein was eluted with a 30–500 mM linear imidazole gradient in the equilibration buffer. Fractions containing RNase H2 were dialyzed overnight in Mono-Q equilibrium buffer (20 mM TrisHCl pH 7.5, 1 mM DTT, 1 mM EDTA, 50 mM NaCl, 10% glycerol) and loaded on a Mono-Q column (GE Healthcare) equilibrated with the same buffer. The protein was then eluted with a NaCl 50 mM-1M linear gradient. Positive fractions were collected and dialyzed for 4 h in 40 mM TrisHCl pH 7.5, 50 mM NaCl and 50% glycerol. Lastly, the enzyme was dialyzed In 40 mM TrisHCl pH 8.5, 125 mM NaCl and 50% glycerol for an additional 4 h.

### 4.3. RNase H2 Assay and Sequence Analysis

The enzymatic activity of RNase H2 was monitored using all DNA substrates listed in Table 1. For the experiments, RNase H2 and DNA were mixed at a final concentration of 0.4 nM (enzyme) and 20 nM (substrate). All reactions were performed in a 10 μL final volume in the following reaction buffer: 20 mM TrisHCl pH 8.5, 60 mM NaCl, 50 g/mL BSA, 1 mM DTT, 1% glycerol, 10 mM MgCl_2_. Reactions were incubated at 37 °C (timing specified in the figure legends) and were stopped by the addition of standard denaturing gel loading buffer (95% formamide, 10 mM ethylenediaminetetraacetic acid, xylene cyanol, and bromophenol blue), heated at 100 °C for 10 min and loaded on a 7 M urea 12% polyacrylamide (PA) gel. The reaction products were acquired with Typhoon Trio (GE Healthcare) and bands were quantified using ImageJ. For each lane, enzyme activity was calculated as the percentage of the product signal on the total signal (product band plus unreacted bands). Data obtained were plotted in a GraphPad Prism 6.0 according to a one-phase association equation. Single-stranded trimming activity was excluded evaluating RNaseH2 prep against ssDNA/RNA hybrid oligonucleotide (Appendix A). All reactions were performed in triplicates.

## Figures and Tables

**Figure 1 ijms-22-05201-f001:**
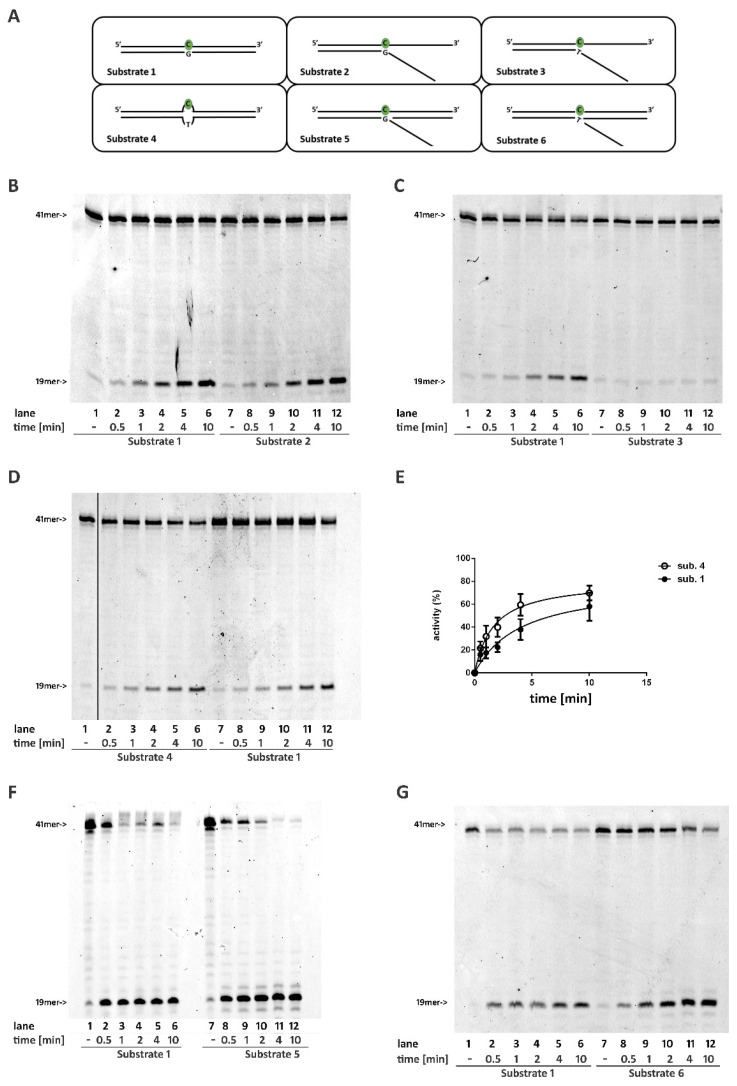
Base pairing at the 3′ of an embedded ribonucleotide is not required for RNaseH2 activity. Time course reactions were performed as described in materials and methods using 20 nM nucleic acid substrates and 0,4 nM RNase H2. (**A**) Schematic representation of substrates used. Enzyme activity was compared between (**B**) Substrates 1 and 2; (**C**) Substrates 1 and 3; (**D**) Substrates 1 and 4; (**F**) Substrates 1 and 5; (**G**) Substrates 1 and 6. Unprocessed substrates and reaction products are indicated at the left of each panel (41-mer and 19-mer respectively). Black vertical line in (**D**) represents the cropped area in the raw picture. (**E**) Quantification of three independent experiments measuring RNase H2 activity using substrate 1 and 4.

**Figure 2 ijms-22-05201-f002:**
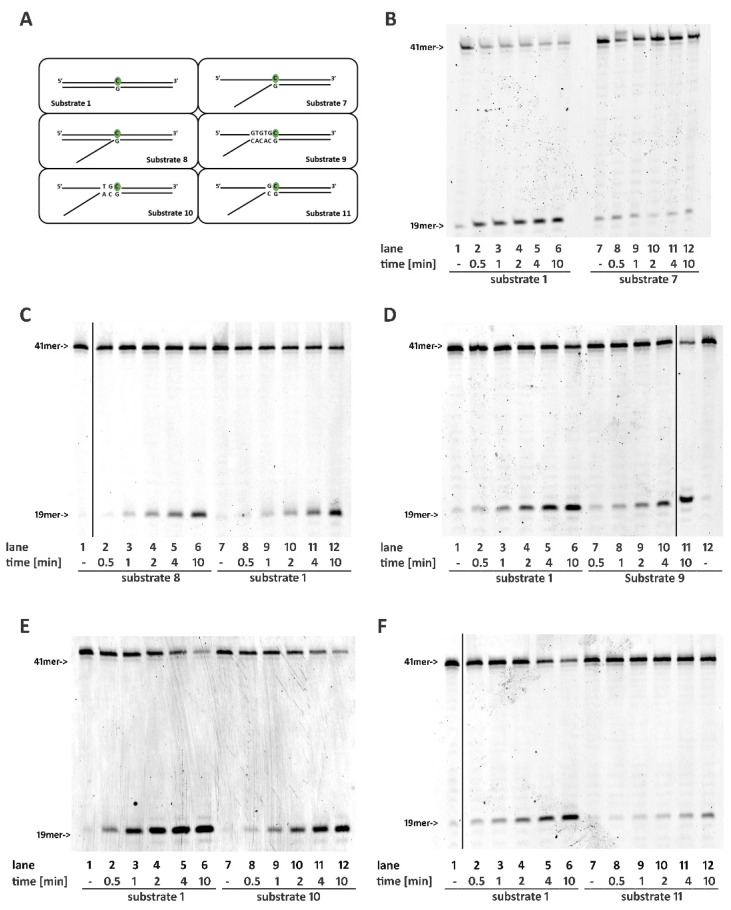
Base pairing at the 5′ of an embedded ribonucleotide is essential for RNase H2 activity. Time course reactions were performed as described in materials and methods using 20 nM nucleic acid substrates and 0.4 nM RNase H2. (**A**) Schematic representation of substrates used. Enzyme activity was compared between (**B**) Substrates 1 and 7; (**C**) Substrates 1 and 8; (**D**) Substrates 1 and 9; (**E**) Substrates 1 and 10; (**F**) Substrates 1 and 11. Unprocessed substrates and reaction products are indicated at the left of each panel (41-mer and 19-mer respectively). Black vertical lines represent cropped areas in the raw pictures.

**Figure 3 ijms-22-05201-f003:**
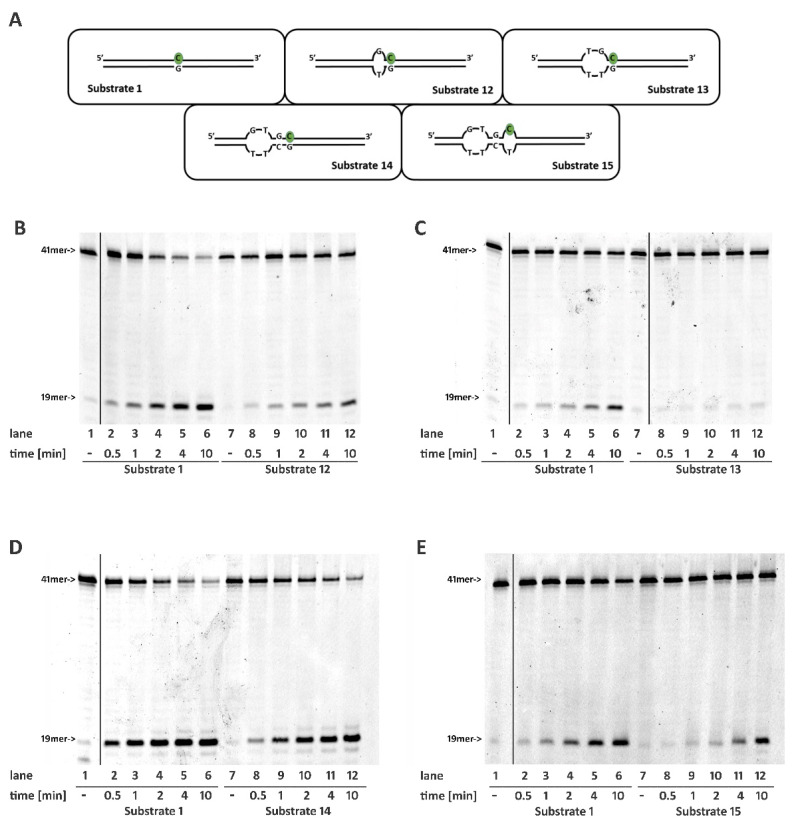
Correct pairing of the two nucleotides flanking the 5′ of the embedded ribonucleotide is essential for RNase H2 activity. Time course reactions were performed as described in materials and methods using 20 nM nucleic acid substrates and 0.4 nM RNaseH2. (**A**) Schematic representation of substrates used. Enzyme activity was compared between (**B**) Substrates 1 and 12; (**C**) Substrates 1 and 13; (**D**) Substrates 1 and 14; (**E**) Substrates 1 and 15. Unprocessed substrates and reaction products are indicated at the left of each panel (41mer and 19mer respectively). Black vertical lines represent cropped areas in the raw pictures.

**Figure 4 ijms-22-05201-f004:**
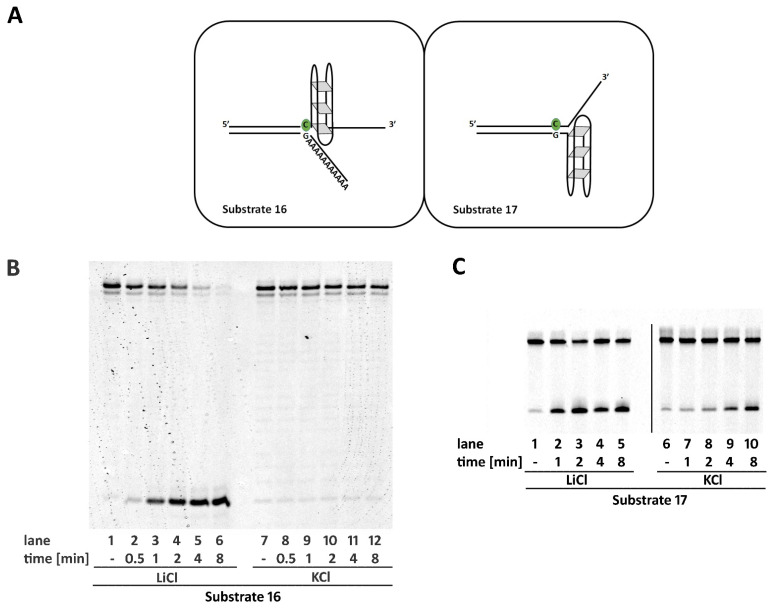
A G4 structure on the same strand and flanking the 3′ of a single ribonucleotide impairs RNase H2 activity. (**A**) schematic representation of Substrates **16** and **17** annealed in the presence of KCl. Time course reactions were performed as described in materials and methods using 20 nM of Substrate **16** (**B**), or **17** (**C**), annealed in the presence of KCl (allowing G4 formation), or in the presence of LiCl (not allowing G4 formation). Unprocessed substrates and reaction products are indicated at the left of each panel (44mer and 22mer respectively).

**Figure 5 ijms-22-05201-f005:**
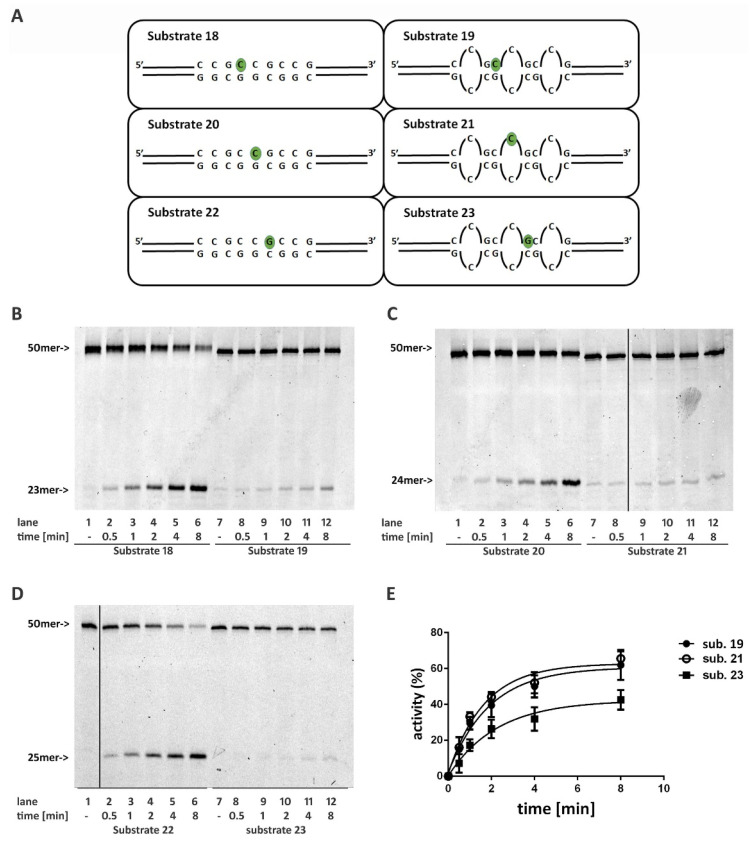
Ribonucleotides embedded in structures mimicking the stem part of CCG hairpins are poor substrates of RNase H2. (**A**) Schematic representation of substrates mimicking the stem of CCG hairpin structures. Time course reactions were performed using 20 nM nucleic acid substrates and 0.4 nM RNase H2. Enzyme activity was compared between (**B**) Substrates **18** and **19**; (**C**) Substrates **20** and **21**; (**D**) Substrates 22 and 23. (**E**) Quantification of three independent RNase H2 reactions using Substrates **19**, **21** and **23**. Unprocessed substrates and reaction products are indicated at the left of each panel (50mer and 25mer respectively). Black vertical lines in (**C**,**D**) represents cropped areas in the raw pictures.

**Figure 6 ijms-22-05201-f006:**
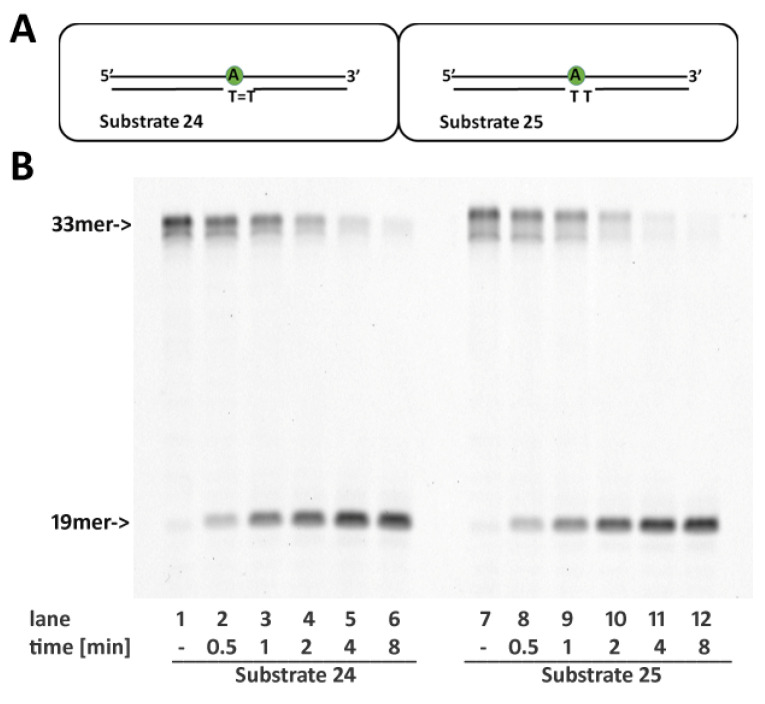
The presence of a CPD lesion opposite a single ribonucleotide does not affects RNase H2 activity. (**A**) Schematic representation of Substrates **24** (CTD) and **25** (TT). (**B**) Enzyme activity was compared between Substrates **24** and **25**. Time course reactions were performed as described in materials and methods using 0.4 nM RNase H2 and 20 nM Substrates **24** and **25** as indicated.

**Table 1 ijms-22-05201-t001:** Substrates and oligonucleotides sequences.

Substrate	Oligo	Sequence
1	#1	5′-Fam-GAGTTTGAGTGGAGGTGTG(rC)GTAGGGTAGTATTGGTGGATA-3′
	#2	5′-TATCCACCAATACTACCCTACGCACACCTCCACTCAAACTC-3′
2	#3	5′-Fam-GAGTTTGAGTGGAGGTGTG(rC)GTAGGGTAGTATTGGTGGATA-3′
	#4	5′-TGAGGAGTTGGTTGCACACCTCCACTCAAACTC-3′
3	#1	Fam
	#5	5′-TGAGGAGTTGGTTTCACACCTCCACTCAAACTC-3′
4	#1	5′-Fam-GAGTTTGAGTGGAGGTGTG(rC)GTAGGGTAGTATTGGTGGATA-3′
	#6	5′-TGAGGAGTTGGTTTCACACCTCCACTCAAACTC-3′
5	#1	5′-Fam-GAGTTTGAGTGGAGGTGTG(rC)GTAGGGTAGTATTGGTGGATA-3′
	#5	5′-TGAGGAGTTGGTTGCACACCTCCACTCAAACTC-3′
	#7	5′-TATCCACCAATACTACCCTAC-3′
6	#1	5′-Fam-GAGTTTGAGTGGAGGTGTG(rC)GTAGGGTAGTATTGGTGGATA-3′
	#6	5′-TGAGGAGTTGGTTTCACACCTCCACTCAAACTC-3′
	#7	5′-TATCCACCAATACTACCCTAC-3′
7	#1	5′-Fam-GAGTTTGAGTGGAGGTGTG(rC)GTAGGGTAGTATTGGTGGATA-3′
	#8	5′-TATCCACCAATACTACCCTACGTTTTGGATGTGGGTTT-3′
8	#1	5′-Fam-GAGTTTGAGTGGAGGTGTG(rC)GTAGGGTAGTATTGGTGGATA-3′
	#8	5′-TATCCACCAATACTACCCTACGTTTTGGATGTGGGTTT-3′
	#9	5′-CACACCTCCACTCAAACTC-3′
9	#1	5′-Fam-GAGTTTGAGTGGAGGTGTG(rC)GTAGGGTAGTATTGGTGGATA-3′
	#10	5′-TATCCACCAATACTACCCTACGTTTTGGATGTGGGTTT-3′
10	#1	5′-Fam-GAGTTTGAGTGGAGGTGTG(rC)GTAGGGTAGTATTGGTGGATA-3′
	#11	5′-TATCCACCAATACTACCCTACGCTTTGGATGTGGGTTT-3′
11	#1	5′-Fam-GAGTTTGAGTGGAGGTGTG(rC)GTAGGGTAGTATTGGTGGATA-3′
	#12	5′-TATCCACCAATACTACCCTACGCATTGGATGTGGGTTT-3′
12	#1	5′-Fam-GAGTTTGAGTGGAGGTGTG(rC)GTAGGGTAGTATTGGTGGATA-3′
	#13	5′-TATCCACCAATACTACCCTACGTACACCTCCACTCAAACTC-3′
13	#1	5′-Fam-GAGTTTGAGTGGAGGTGTG(rC)GTAGGGTAGTATTGGTGGATA-3′
	#14	5′-TATCCACCAATACTACCCTACGTTCACCTCCACTCAAACTC-3′
14	#1	5′-Fam-GAGTTTGAGTGGAGGTGTG(rC)GTAGGGTAGTATTGGTGGATA-3′
	#15	5′-TATCCACCAATACTACCCTACGCTGACCTCCACTCAAACTC-3′
15	#1	5′-Fam-GAGTTTGAGTGGAGGTGTG(rC)GTAGGGTAGTATTGGTGGATA-3′
	#16	5′-TATCCACCAATACTACCCTACTCTGACCTCCACTCAAACTC-3′
16	#17	5′-Fam-GAGAGAAGATGGAAGTGAGGAA(rC)GGGATAGGGATAGGGATAGGG
	#18	5′-AAAAAAAAAAAAAAAAAAAAAGTTCCTCACTTCCATCTTCTCTC-3′
17	#19	5′-Fam-GAGAGAAGATGGAAGTGAGGAA(rC)AAAAAAAAAAAAAAAAAAAAAAA-3′
	#20	5′-GGGATAGGGATAGGGATAGGGTTGTTCCTCACTTCCATCTTCTCTC-3′
18	#21	5′-Fam-GAGTTTGAGTGTAGTGTTGTCCG(rC)CGCCGGTAGAGTAGTATGTGTTGATA-3′
	#22	5′-TATCAACACATACTACTCTACGCCGCCGCCACAACACTACACTCAAACTC-3′
19	#21	5′-Fam-GAGTTTGAGTGTAGTGTTGTCCG(rC)CGCCGGTAGAGTAGTATGTGTTGATA-3′
	#23	5′-TATCAACACATACTACTCTACCGGCGGCGGACAACACTACACTCAAAC-3′
20	#24	5′-Fam-GAGTTTGAGTGTAGTGTTGTCCGC(rC)GCCGGTAGAGTAGTATGTGTTGATA-3′
	#22	5′-TATCAACACATACTACTCTACGCCGCCGCCACAACACTACACTCAAACTC-3′
21	#24	5′-Fam-GAGTTTGAGTGTAGTGTTGTCCGC(rC)GCCGGTAGAGTAGTATGTGTTGATA-3′
	#23	5′-TATCAACACATACTACTCTACCGGCGGCGGACAACACTACACTCAAAC-3′
22	#25	5′-Fam-GAGTTTGAGTGTAGTGTTGTCCGCC(rG)CCGGTAGAGTAGTATGTGTTGATA-3′
	#22	5′-TATCAACACATACTACTCTACGCCGCCGCCACAACACTACACTCAAACTC-3′
23	#25	5′-Fam-GAGTTTGAGTGTAGTGTTGTCCGCC(rG)CCGGTAGAGTAGTATGTGTTGATA-3′
	#23	5′-TATCAACACATACTACTCTACCGGCGGCGGACAACACTACACTCAAAC-3′
24	#26	5′-Fam-GGGTGGAGGTGACA[rA]GAGTGTGATAGAGTGTGA-3′
	#27	5′-TCACACTCTATCACACTC[CPD]GTCACCTCCACCC
25	#26	5′-Fam-GGGTGGAGGTGACA[rA]GAGTGTGATAGAGTGTGA-3′
	#28	5′-TCACACTCTATCACACTCTTGTCACCTCCACCC

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
