# Peer review of "High Flexibility of RNaseH2 Catalytic Activity with Respect to Non-Canonical DNA Structures"

_ijms, 2021, doi:10.3390/ijms22105201_

Round 1

Reviewer 1 Report

Review of ijms-1215578

April 26, 2021

Dede et al. extends previous studies on RNaseH2. Their previous observation was that an rN paired with Cis-Pt in DNA is a poor substrate for RNaseH2 due to distorted geometry. They extended this observation to examine RNaseH2 activity in the context of a variety of biologically relevant DNA structures (flaps, mismatches, G4). The experimental design and analysis is logical, clear and maps important determinants for RNaseH2 activity. The findings contribute to the literature and towards a more comprehensive view of RNaseH2 in a variety of contexts. The data in the manuscript help define "rules" for RNaseH2 substrate cleavage. The Discussion effectively puts the data into structure-function (discussion of known structural models) and biological context (relationship to repeats and TLS pols). I support publishing this manuscript with minor revision.

Minor comments:

1. A important controls is RNaseH2 activity on ssDNA containing a single rN. Please include this control.

2. Please increase the size of the labels and nt in the figures to increase readibility.

3. Typos:

            Line 148: "observe" to "observed"

            Line 300: "Coli" to "coli"

            Line 334: "bais" to "base"

            Line 399: "Coli" to "coli"

Author Response

We thank the reviewer for her/his suggestion.

Point 1.  A important controls is RNaseH2 activity on ssDNA containing a single rN. Please include this control.

Response 1. The control required was included (see supplementary Figure S8). As expected, RNase H2 shows no activity against ssDNA containing a single rN.

Point 2. Please increase the size of the labels and nt in the figures to increase readibility.

Response 2. Figures size was increased making labels and nt easier to read.

Point 3. Typos: 
Line 148: "observe" to "observed"
Line 300: "Coli" to "coli"
Line 334: "bais" to "base"
Line 399: "Coli" to "coli"

Response 3. Done

Reviewer 2 Report

In this manuscript Dede et al. describe a thorough biochemical characterisation of RNaseH2 activity in the presence of a series of different DNA substrates containing ribonucleotide lesions. It is a fine study with important potential implications in the understanding of the fundamental mechanisms linking RNaseH2 activity, with respect to different potential sites of ribonulceotide incorporation, and genomic-instability events. In light of this I am overall supportive for publication. There are however some improvements that I would recommend prior publication, most of those have been already addressed by the authors in this revised version of the manuscript that I have previously reviewed when submitted somewhere else.

1) The minor improvement that is still required would be to specify explicitly in the text the G4-sequences used in the study, so that it will be more immediate to pick up what are the G4s tested without going necessarily to the supporting information. Also it would be worth spelling out in the main text that multiple G4 sequences have been tested, currently it still reads like one one sequence has been used, which does not reflect the extend of the additional work that the authors have performed. 

Subject to the above I am fully supportive of publication

Author Response

We thank the reviewer for her comments.

Point 1. The minor improvement that is still required would be to specify explicitly in the text the G4-sequences used in the study, so that it will be more immediate to pick up what are the G4s tested without going necessarily to the supporting information. Also it would be worth spelling out in the main text that multiple G4 sequences have been tested, currently it still reads like one one sequence has been used, which does not reflect the extend of the additional work that the authors have performed.

Response 1. We explicit the sequences used within the text as required. In the text was already mentioned that we used two different G4 sequences. We put more emphasis on this rearranging the sequence as “… we decided to evaluate the activity of the enzyme on substrates presenting G4 structures composed by two different sequences at the 3’ of a single embedded ribonucleotide, both known to form stable G4 structures (GGGATAGGGATAGGGATAGGG sequence within Substrates 16, and cmyc22 sequence – TGAGGGTGGGTAGGGTGGG-TAA [18] within Substrate S3).